# SFTMix: Elevating Language Model Instruction Tuning with Mixup Recipe

## Abstract

To induce desired behaviors in large language models (LLMs) for interaction-driven tasks, the instruction-tuning stage typically trains LLMs on instruction-response pairs using the next-token prediction (NTP) loss. Previous work aiming to improve instruction-tuning performance often emphasizes the need for higher-quality supervised fine-tuning (SFT) datasets, which typically involves expensive data filtering with proprietary LLMs or labor-intensive data generation by human annotators. However, these approaches do not fully leverage the datasets' intrinsic properties, resulting in high computational and labor costs, thereby limiting scalability and performance gains. In this paper, we propose SFTMix, a novel recipe that elevates instruction-tuning performance beyond the conventional NTP paradigm, without the need for well-curated datasets. Observing that LLMs exhibit uneven confidence across the semantic representation space, we argue that examples with different confidence levels should play distinct roles during the instruction-tuning process. Based on this insight, SFTMix leverages training dynamics to identify examples with varying confidence levels, then applies a Mixup-based regularization to mitigate overfitting on confident examples while propagating supervision signals to improve learning on relatively unconfident ones. This approach enables SFTMix to significantly outperform NTP across a wide range of instruction-following and healthcare domain-specific SFT tasks, demonstrating its adaptability to diverse LLM families and scalability to datasets of any size. Comprehensive ablation studies further verify the robustness of SFTMix's design choices, underscoring its versatility in consistently enhancing performance across different LLMs and datasets in broader natural language processing applications.[1]

## 1 Introduction

Large language models (LLMs) have recently demonstrated outstanding performance across a broad spectrum of natural language processing (NLP) tasks (Zhao et al., 2023; Minaee et al., 2024). After being pre-trained on large corpora of raw text, LLMs undergo a critical instruction-tuning stage (Ouyang et al., 2022; Zhang et al., 2023) to develop their instruction-following capabilities based on supervised fine-tuning (SFT) datasets, making them more suitable for interaction-driven applications. SFT datasets (Taori et al., 2023; Wang et al., 2023; Xu et al., 2024) typically consist of instruction-response pairs spanning various task types, aligning LLMs toward desired behavior. During this stage, LLMs are usually trained through next-token prediction (NTP), where LLMs predict the next token in a response given both the instruction and the preceding tokens in that response.

Previous research efforts in this field have predominantly focused on enhancing the quality of instruction-tuning datasets. One line of research direction seeks to better understand the intrinsic properties of these datasets (Kung et al., 2023; Lin et al., 2024) and selects informative instruction-response pairs through heuristics-based filters (Zhao et al., 2024) or LLM scoring (Chen et al., 2024). Another line of work generates high-quality responses by querying advanced proprietary LLMs (Chen et al., 2024) or relying on human annotators (Zhou et al., 2023). However, both strategies come with significant computational or labor costs, limiting the scalability of SFT datasets.

In this paper, we take a different perspective by exploring how to elevate instruction-tuning performance beyond the conventional NTP training paradigm, without relying on well-curated datasets

---

[1]We will release our implementation after the anonymous review period.

To address this challenge, we propose SFTMix, a novel Mixup-based recipe for language model instruction tuning. Our design builds upon the key observation that an LLM's confidence distribution over its instruction-tuning dataset is uneven across the semantic representation space. We argue that data with varying confidence levels should contribute differently during the instruction-tuning process. Hence, we extend data cartography (Swayamdipta et al., 2020) to the realm of causal language generation as training dynamics and leverage a reference LLM to derive the confidence of each instruction-response pair, based on the perplexities computed over the instruction-tuning process. Using this information, we divide the original SFT dataset into a confident subset and a relatively unconfident subset of equal size.

To guide the learning of the unconfident examples by propagating supervision signals, and to mitigate overfitting to the confident semantic regions, we design a Mixup-based (Zhang et al., 2018) regularization for LLM instruction tuning. This approach explores the utility of Mixup in causal language generation and improves its effectiveness by exploiting the confidence information derived from training dynamics. Specifically, consider an instruction-response pair from the confident subset and another from the unconfident subset. We interpolate them linearly at the token level in the representation space and generate a convex combination of the representations from the LLM under instruction tuning. The one-hot encodings of the corresponding tokens are interpolated in a similar fashion. We then compute a regularization between the interpolated encodings and representations, in addition to the original NTP loss during instruction tuning. In this way, SFTMix fosters the synergy between the two subsets with diverging confidence levels and enhances the interaction capabilities of LLMs across diverse downstream applications.

We demonstrate the effectiveness of our proposed SFTMix recipe in both instruction-following and domain-specific SFT settings. In particular, SFTMix significantly surpasses the conventional NTP instruction-tuning baseline in both single- and multi-turn conversations, as measured in MT-Bench (Zheng et al., 2024) and AlpacaEval-2 (Dubois et al., 2024). This improvement is consistent across different LLM families (e.g., Llama (Dubey et al., 2024) and Mistral (Jiang et al., 2023)) and scales of SFT datasets (e.g., Alpaca-52K (Taori et al., 2023) and UltraChat-200K (Tunstall et al., 2023)). Moreover, in the healthcare domain, Llama-3.1-8B (Dubey et al., 2024) and Mistral-7B-v0.1 (Jiang et al., 2023), instruction-tuned on MedAlpaca-263K (Han et al., 2023) using SFTMix, achieve an average of $1.5\%$ absolute increase in accuracy across four benchmarks compared to baselines. We further explore the applicability of SFTMix's variants through extensive ablations and illustrate the potential of this LLM instruction-tuning recipe for broader use cases.

We summarize our contributions in this paper as follows:

- We introduce SFTMix, a novel recipe designed to elevate LLM instruction-tuning performance beyond the conventional NTP paradigm, without relying on well-curated datasets.

- Motivated by the observation that LLMs exhibit varying confidence levels across the semantic space, SFTMix leverages LLMs' training dynamics for more insightful data interpretation and confidence-based splitting, facilitating more effective instruction tuning.

- SFTMix further incorporates a Mixup-based regularization that interpolates between examples with different confidence levels during instruction tuning, mitigating overfitting to confident examples while improving generalization on relatively unconfident ones.

- We demonstrate that SFTMix significantly outperforms the NTP baseline across a variety of instruction-following and healthcare domain-specific SFT tasks, with consistent improvements across different LLM families and dataset sizes.

- Comprehensive ablation analysis substantiates the robustness of our design choices in SFTMix, shedding light on its potential for broader NLP applications.

## 2 RELATED WORK

**LLM Instruction Tuning.** To align LLMs with users' open-ended intents or adapt them to specific domains, Ouyang et al. (2022) proposed instruction-tuning LLMs on human-annotated demonstrations using supervised learning. More specifically, given a pair of instructions and desired responses, the conventional NTP paradigm trains an LLM to predict each token in the response sequentially during the instruction-tuning stage (Zhang et al., 2023). Jain et al. (2024) improved instruction-tuning performance by adding noise to the token embeddings during training, while Shi et al. (2024)

further suggested modeling the instructions as well. On this basis, previous work (Chiang et al., 2023; Ding et al., 2023; Taori et al., 2023; Wang et al., 2023; Xu et al., 2024) collected instruction-following datasets by distilling powerful proprietary LLMs or crowdsourcing user conversations. To enhance data quality, the community has employed various techniques, including heuristic-based filters (Schoch et al., 2023; Zhao et al., 2024), LLM scoring (Chen et al., 2024), and human curation (Zhou et al., 2023). Other efforts (Kung et al., 2023; Lin et al., 2024) have focused on gaining a deeper understanding of the intrinsic properties of SFT datasets. However, acquiring high-quality SFT data often entails substantial computational and labor costs. In this paper, we aim to optimize data utilization through insightful data interpretation and improve the effectiveness of instruction tuning beyond the conventional NTP paradigm, without relying on well-curated datasets.

**Data Characterization via Training Dynamics.** Data characterization (Albalak et al., 2024; Wang et al., 2024) seeks to assess and analyze the quality and relevance of training data, enabling more effective data filtering and elevated model performance. In particular, Swayamdipta et al. (2020) leveraged the training dynamics of a pre-trained language model (Liu, 2019) to create data maps, which have subsequently inspired advancements in active learning (Zhang & Plank, 2021; Zhang et al., 2022; Kung et al., 2023), curriculum learning (Christopoulou et al., 2022; Lin et al., 2024; Poesina et al., 2024), and dataset pruning (Chimoto et al., 2024; He et al., 2024; Lin et al., 2024; Seedat et al., 2024). Here, we explore applying training dynamics to causal language generation by categorizing an SFT dataset into confident and relatively unconfident subsets, which facilitates the subsequent Mixup-based regularization during LLM instruction tuning.

**Mixup-Based Learning.** To alleviate memorization and sensitivity to adversarial examples during training, Zhang et al. (2018) proposed Mixup, which trains models on convex combinations of pairs of input features and their corresponding labels. Its variants (Verma et al., 2019; Hendrycks et al., 2020; Uddin et al., 2021; Choi et al., 2022) further suggest interpolating feature representations at different stages, guided by various training signals. Theoretical analyses (Zhang et al., 2021; Carratino et al., 2022; Chidambaram et al., 2022; Park et al., 2022; Pinto et al., 2022) have demonstrated its data-adaptive regularization and generalization effects, leading to strong out-of-distribution robustness and well-calibrated uncertainty estimation. Empirical studies have further validated its effectiveness under the semi-supervised learning setting (Berthelot et al., 2019; 2020; Li et al., 2020; 2022) and in diverse NLP applications (Chen et al., 2020; Guo et al., 2020; Sun et al., 2020; Park & Caragea, 2022; Yang et al., 2022). Building on this success, we explore its utility in LLM instruction tuning and propose a Mixup-based regularization to reduce overfitting to confident examples and support the learning of relatively unconfident ones.

## 3 SFTMIX

In Section 3.1, we begin by reviewing the conventional instruction-tuning task of NTP. We then introduce SFTMix, a novel recipe for LLM instruction tuning. SFTMix first leverages training dynamics to determine subspaces with distinct confidence levels (Section 3.2). Subsequently, it incorporates a Mixup-based regularization (Section 3.3) to mitigate overfitting to confident examples and propagate their supervision signals to promote the learning of relatively unconfident ones. We illustrate the overall pipeline of SFTMix in Figure 1.

### 3.1 PRELIMINARIES OF THE CONVENTIONAL NTP INSTRUCTION-TUNING PARADIGM

Consider an SFT dataset $\mathcal{D} = \{(\mathcal{X}_i, \mathcal{Y}_i)\}_{i=1}^{|\mathcal{D}|}$, consisting of pairs of instructions $\mathcal{X}_i$ and desired responses $\mathcal{Y}_i$. Here, both $\mathcal{X}_i$ and $\mathcal{Y}_i$ are sequences of tokens, where $\mathcal{X}_i = (x_1, \ldots, x_{M_i})$ and $\mathcal{Y}_i = (y_1, \ldots, y_{N_i})$. The conventional NTP task minimizes the following loss for predicting $\mathcal{Y}_i$ given $\mathcal{X}_i$:

$$\ell_{\text{NTP}}(\mathcal{D}) = -\sum_{i=1}^{|\mathcal{D}|} \sum_{n=1}^{N_i} \log p(y_n \mid \mathcal{X}_i, y_1, \ldots, y_{n-1}) = -\sum_{i=1}^{|\mathcal{D}|} \sum_{n=1}^{N_i} H(\mathbf{Y}_n, \sigma(\mathbf{Z}_n)). \quad (1)$$

This loss equals the sum of negative cross-entropy $H$ between $\mathbf{Y}_n$ and $\mathbf{Z}_n$ after softmax $\sigma$, where $\mathbf{Y}_n$ is the one-hot encoding of the $n$-th token in $\mathcal{Y}_i$, and $\mathbf{Z}_n = \text{LLM}(\mathcal{X}_i, y_1, \ldots, y_{n-1})$ is the corresponding representation generated by the LLM's causal language modeling head.

To improve the effectiveness of SFT, existing work (Zhou et al., 2023; Chen et al., 2024) has primarily focused on providing higher-quality SFT datasets. However, these approaches often lack

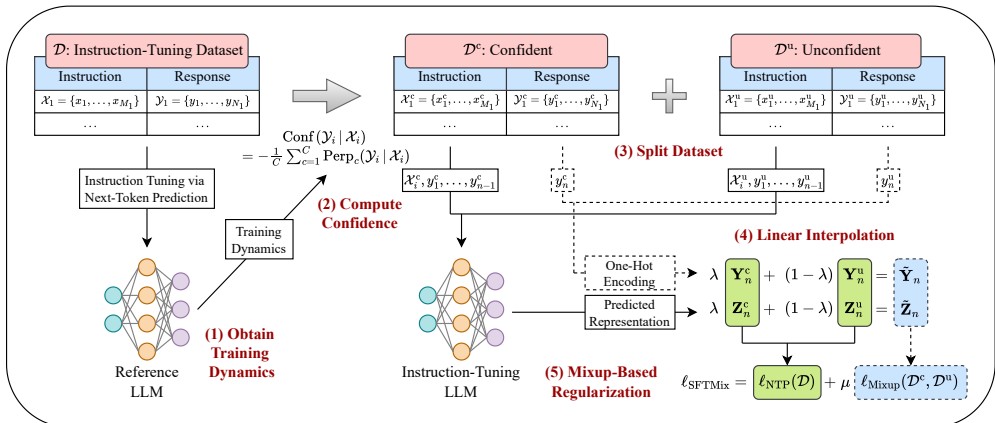

Figure 1: The overall pipeline of the proposed SFTMix recipe for LLM instruction tuning. Given an SFT dataset $\mathcal{D}$, we (1) train a reference LLM on $\mathcal{D}$ using NTP and (2) compute $\mathrm{Conf}(\mathcal{Y}_i \,|\, \mathcal{X}_i)$ for $(\mathcal{X}_i, \mathcal{Y}_i) \in \mathcal{D}$ based on the training dynamics of the reference LLM. On this basis, we (3) divide $\mathcal{D}$ into a confident subset $\mathcal{D}^{\mathrm{c}}$ and a relatively unconfident one $\mathcal{D}^{\mathrm{u}}$ of equal size. Finally, given pairs of examples from each subset, we (4) interpolate their one-hot encodings and predicted representations linearly at the token level and (5) incorporate a Mixup-based regularization $\ell_{\mathrm{Mixup}}(\mathcal{D}^{\mathrm{c}}, \mathcal{D}^{\mathrm{u}})$ in addition to the NTP loss $\ell_{\mathrm{NTP}}(\mathcal{D})$ during LLM instruction tuning.

an insightful understanding of SFT datasets and incur significant computational and labor costs, limiting scalability and performance gains. In response, we propose SFTMix, a novel instruction-tuning recipe in the following sections, which enhances instruction-tuning performance beyond the conventional NTP paradigm without relying on well-curated datasets.

## 3.2 TRAINING DYNAMICS IDENTIFY SUBSETS WITH DISTINCT CONFIDENCE LEVELS

Suppose we identify $C$ checkpoints of a reference LLM when instruction-tuning it using the NTP task in Section 3.1. We aim to capture the training dynamics of the reference LLM by computing its confidence in generating each pair $(\mathcal{X}_i, \mathcal{Y}_i) \in \mathcal{D}$. More specifically, we define confidence based on the perplexity of $\mathcal{Y}_i$ given $\mathcal{X}_i$ at each checkpoint $c \in \{1, \ldots, C\}$:

$$\mathrm{Perp}_c(\mathcal{Y}_i \,|\, \mathcal{X}_i) = -\frac{1}{N_i} \sum_{n=1}^{N_i} \log p(y_n \,|\, \mathcal{X}_i, y_1, \ldots, y_{n-1}) = -\frac{1}{N_i} \sum_{n=1}^{N_i} \mathbf{Y}_n^\top \log \sigma(\mathbf{Z}_n), \qquad (2)$$

$$\mathrm{Conf}(\mathcal{Y}_i \,|\, \mathcal{X}_i) = -\frac{1}{C} \sum_{c=1}^{C} \mathrm{Perp}_c(\mathcal{Y}_i \,|\, \mathcal{X}_i). \qquad (3)$$

Note that $\mathbf{Z}_n$ here is produced by the reference LLM at checkpoint $c$. The reference LLM's confidence in predicting $\mathcal{Y}_i$ given $\mathcal{X}_i$ is the negative average perplexity over the $C$ checkpoints, as a lower perplexity indicates a higher likelihood of generation.

**LLMs Exhibit Uneven Confidence across the Semantic Representation Space.** Here, we present a case study by instruction-tuning Llama-3.1-8B (Dubey et al., 2024) on Alpaca-52K (Taori et al., 2023) and collect the LLM's confidence for each training data point across five checkpoints. We use the last hidden state of the final token in $(\mathcal{X}_i, \mathcal{Y}_i)$ as its embedding and plot 2,500 high-confidence and 2,500 low-confidence data points in Figure 2 (a) via t-SNE (Van der Maaten & Hinton, 2008). Correspondingly, we present one confident and one unconfident example from these data points in Figure 2 (b). We observe that embeddings of data points with contrasting confidence levels are clearly separated in Figure 2 (a), indicating that the distribution of the LLM's confidence is uneven across the semantic representation space. This observation is further supported by the examples in Figure 2 (b), where the LLM exhibits high confidence in the example discussing deterministic grammar rules and low confidence in the example concerning creative content in e-commerce.

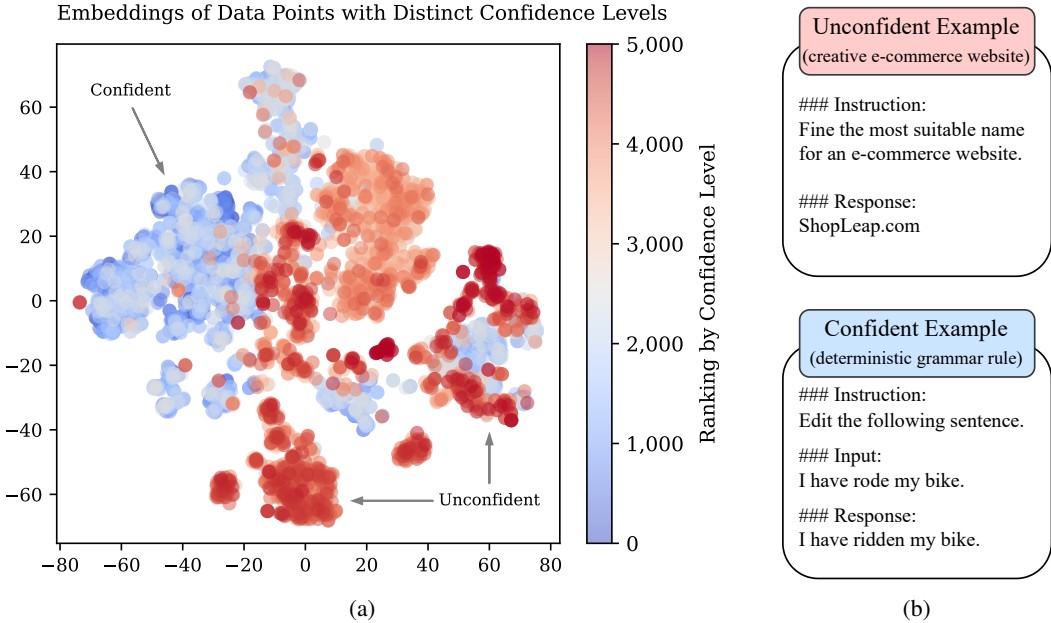

Figure 2: (a) Embeddings of 2,500 high-confidence and 2,500 low-confidence examples in Alpaca-52K by Llama-3.1-8B trained on Alpaca-52k using NTP. (b) Corresponding confident and unconfident examples. The clear separation between embeddings of high-confidence and low-confidence examples suggests that the LLM exhibits varying confidence levels across different semantic regions, as further illustrated by the different topics in the provided examples.

**Data with Distinct Confidence Levels Should Play Different Roles during Instruction Tuning.**
The insight from the case study motivates us to contend that data with varying confidence levels should contribute differently during instruction tuning. Highly confident data points typically lie further from the classification decision boundary, posing a higher risk of overfitting. In contrast, less confident data points are often closer to the boundary, making them harder to learn. To address this, we propose promoting the flow of supervision signals between confident and less confident regions to mitigate overfitting and enhance generalization during LLM instruction tuning. On this basis, we divide the original SFT dataset $\mathcal{D}$ into a confident subset $\mathcal{D}^{\mathrm{c}}$ and a relatively unconfident subset $\mathcal{D}^{\mathrm{u}}$ of equal size according to $\mathrm{Conf}\,(\mathcal{Y}_i \mid \mathcal{X}_i)$. To foster synergy between them, we design a Mixup-based regularization tailored to the specific challenges of instruction tuning, detailed in the next section.

## 3.3 A MIXUP-BASED REGULARIZATION FACILITATES LLM INSTRUCTION TUNING

To instruction-tune an LLM (different from the reference LLM used to obtain learning dynamics) with our SFTMix recipe, we introduce a novel regularization $\ell_{\mathrm{Mixup}}$ in addition to the conventional NTP loss $\ell_{\mathrm{NTP}}$. Specifically, consider a confident instruction-response pair $(\mathcal{X}_i^{\mathrm{c}}, \mathcal{Y}_i^{\mathrm{c}}) \in \mathcal{D}^{\mathrm{c}}$ and a relatively unconfident pair $(\mathcal{X}_i^{\mathrm{u}}, \mathcal{Y}_i^{\mathrm{u}}) \in \mathcal{D}^{\mathrm{u}}$. Let $\mathbf{Y}_n^{\mathrm{c}}$ and $\mathbf{Y}_n^{\mathrm{u}}$ be the one-hot encoding vectors of the $n$-th token in $\mathcal{Y}^{\mathrm{c}}$ and $\mathcal{Y}^{\mathrm{u}}$, respectively, with $\mathbf{Z}_n^{\mathrm{c}}$ and $\mathbf{Z}_n^{\mathrm{u}}$ as the corresponding representations predicted by the instruction-tuning LLM. We linearly interpolate the two pairs as follows:

$$\tilde{\mathbf{Z}}_n = \lambda \mathbf{Z}_n^{\mathrm{c}} + (1 - \lambda)\mathbf{Z}_n^{\mathrm{u}}, \;\; \tilde{\mathbf{Y}}_n = \lambda \mathbf{Y}_n^{\mathrm{c}} + (1 - \lambda)\mathbf{Y}_n^{\mathrm{u}}, \tag{4}$$

where $\lambda \sim \mathrm{Beta}(\alpha, \alpha)$ and $\alpha$ is a hyperparameter. Suppose that $N_i' = \min(N_i^{\mathrm{c}}, N_i^{\mathrm{u}})$ represents the length of the shorter response between $\mathcal{Y}_i^{\mathrm{c}}$ and $\mathcal{Y}_i^{\mathrm{u}}$. We define the Mixup-based regularization $\ell_{\mathrm{Mixup}}(\mathcal{D}^{\mathrm{c}}, \mathcal{D}^{\mathrm{u}})$ between the confident and relatively unconfident subsets and the overall instruction-tuning loss $\ell_{\mathrm{SFTMix}}$ used in our SFTMix recipe as follows:

$$\ell_{\mathrm{Mixup}}(\mathcal{D}^{\mathrm{c}}, \mathcal{D}^{\mathrm{u}}) = -\sum_{i=1}^{|D|/2} \sum_{n=1}^{N_i'} H(\tilde{\mathbf{Y}}_n, \sigma(\tilde{\mathbf{Z}}_n)), \;\; \ell_{\mathrm{SFTMix}}(\mathcal{D}) = \ell_{\mathrm{NTP}}(\mathcal{D}) + \mu\, \ell_{\mathrm{Mixup}}(\mathcal{D}^{\mathrm{c}}, \mathcal{D}^{\mathrm{u}}). \tag{5}$$

Here, $\mu$ is a hyperparameter to control the regularization effect.

| Dataset | Instruction-Tuning LLM | Recipe | MT-Bench | | | AlpacaEval-2 | |
| | | | Single-Turn | Multi-Turn | Overall | Win Rate | LC Win Rate |
|---------|------------------------|--------|-------------|------------|---------|----------|-------------|
| Alpaca-52K | Llama-3.1-8B | NTP | 4.9100 | 3.8150 | 4.3625 | 4.0714 | 8.6528 |
| | | SFTMix | **5.2125** | **3.9525** | **4.5825** | **4.9031** | **10.3195** |
| | Mistral-7B-v0.1 | NTP | 5.1650 | 4.0675 | 4.6163 | 4.3560 | 9.1759 |
| | | SFTMix | **5.2775** | **4.5425** | **4.9100** | **4.5386** | **9.4994** |
| UltraChat-200K | Llama-3.1-8B | NTP | 6.1875 | 5.0125 | 5.6000 | 5.0665 | 8.4505 |
| | | SFTMix | **6.2750** | **5.3500** | **5.8125** | **5.1149** | **9.3810** |
| | Mistral-7B-v0.1 | NTP | 5.7625 | 4.6938 | 5.2281 | 4.4899 | 7.7732 |
| | | SFTMix | **5.9813** | **4.8813** | **5.4313** | **4.6117** | **8.7650** |

Table 1: Evaluation of instruction-following capabilities of LLMs trained with NTP or SFTMix. We report the average score based on five rounds of evaluation, with the scores from the best-performing instruction-tuning recipe in bold. Standard errors are provided in Appendix A. SFTMix outperforms NTP on both MT-Bench and AlpacaEval-2, irrespective of LLM families and SFT data sizes.

Altogether, as shown in Figure 1, our SFTMix recipe first identifies subspaces with distinct confidence levels using training dynamics, then facilitates the propagation of supervision signals between these subspaces through a Mixup-based regularization. Since LLMs exhibit varying confidence levels across diverse semantic regions, the Mixup-based regularization encourages linear behavior and a smoother decision boundary between confident and unconfident data points (Zhang et al., 2018; Verma et al., 2019). In this way, SFTMix regularizes overfitting in confident regions and propagates supervision signals (Bengio et al., 2009; Chapelle et al., 2009; Sohn et al., 2020) to enhance generalization in less confident regions, thereby improving the instruction-tuning performance.

## 4 EXPERIMENTS

In this section, we assess the effectiveness of SFTMix against the NTP baseline in both instruction-following (Section 4.1) and domain-specific (Section 4.2) SFT tasks. SFTMix consistently improves instruction-tuning performance across different LLM families and SFT datasets of varying scales.

### 4.1 INSTRUCTION-FOLLOWING SFT

Instruction-following SFT trains LLMs on labeled datasets consisting of instructional prompts and corresponding desired responses, enhancing their conversational capabilities in downstream interaction-driven applications. Here, we compare SFTMix with the conventional NTP paradigm by applying them to the instruction tuning of two pre-trained LLMs from different model families (i.e., Llama (Dubey et al., 2024) and Mistral (Jiang et al., 2023)) on two instruction-following datasets of varying scales (i.e., Alpaca-52K (Taori et al., 2023) and UltraChat-200K (Tunstall et al., 2023)). We then evaluate the instruction-tuned LLMs on two widely-adopted benchmarks: MT-Bench (Zheng et al., 2024) and AlpacaEval-2 (Dubois et al., 2024).

**Datasets.** Here, we focus on the following datasets with different sizes.

- Alpaca-52K (Taori et al., 2023) builds on the pipeline from Wang et al. (2023) by prompting text-davinci-003 (Ouyang et al., 2022) to generate diverse instructions and appropriate responses, which results in 52,000 single-turn interactions. We follow its default system prompt and conversation template when preparing training inputs.
- UltraChat-200K (Tunstall et al., 2023) filters out uninformative responses from the original UltraChat dataset (Ding et al., 2023) and downsamples it to 200,000 multi-turn interactions. To adapt these for our SFT pipeline, we expand each multi-turn interaction into multiple single-turn interactions by incorporating the chat history into the instructions.

**Implementation Details.** We experiment with Llama-3.1-8B (Dubey et al., 2024) and Mistral-7B-v0.1 (Jiang et al., 2023) due to their recent release and state-of-the-art performance compared to other models of similar sizes. By default, we use different instances of the same LLM type to obtain training dynamics and for instruction tuning. We determine the instruction-tuning hyperparameters through a coarse sweep on Llama-3.1-8B with Alpaca-52K and adopt them as the default settings in our experiments. Specifically, we train each LLM on Alpaca-52K for three epochs and on UltraChat-200K for one epoch, leveraging eight H100 GPUs. We use the AdamW optimizer with a learning

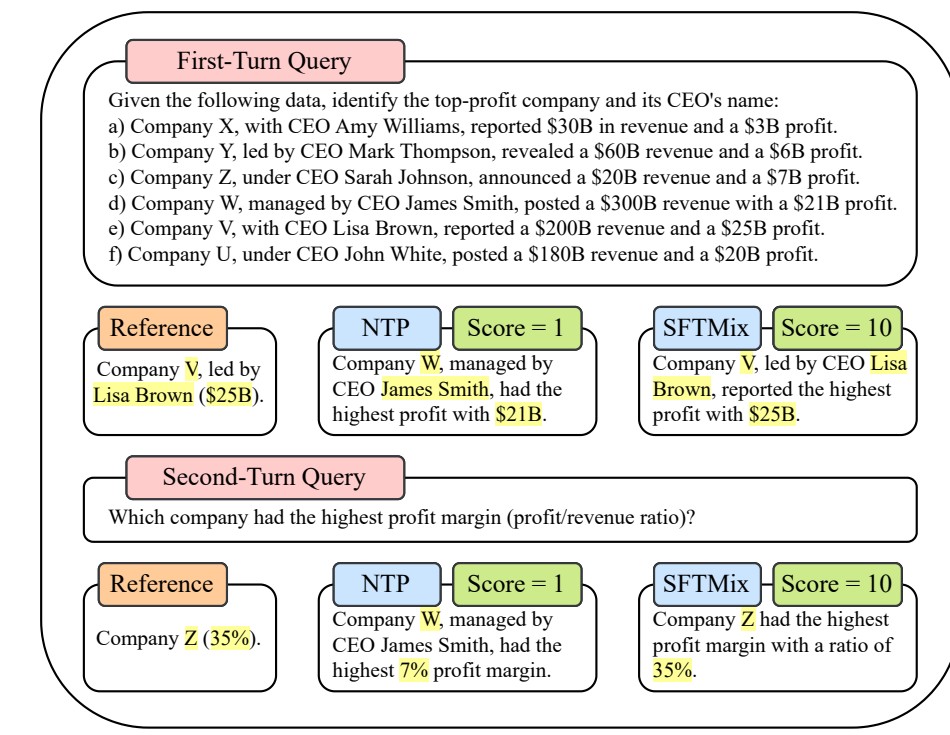

Figure 3: Examples from the extraction category in MT-Bench. Compared to its NTP-tuned counterpart, Llama-3.1-8B instruction-tuned on Alpaca-52K using SFTMix accurately interprets the queries from both turns and correctly extracts the relevant information from the prompt.

rate of $2\mathrm{e}{-6}$ and weight decay of $0.1$, along with a cosine learning rate scheduler featuring a $0.1$ warm-up ratio. Each gradient update during instruction tuning accumulates four batches, with each batch containing eight training examples. We set $\alpha = 0.5$ for sampling $\lambda$ in Equation 4 and $\mu = 0.2$ when constructing $\ell_{\text{SFTMix}}$ in Equation 5. The NTP baseline follows the same instruction-tuning setting without the Mixup-based regularization $\ell_{\text{Mixup}}$ in Equation 5.

**Evaluation Benchmarks.** To evaluate how SFTMix improves LLMs' instruction-following abilities compared to NTP, we assess LLMs instruction-tuned with each method on MT-Bench (Zheng et al., 2024) and AlpacaEval-2 (Li et al., 2023; Dubois et al., 2024). MT-Bench is a challenging benchmark of $80$ multi-turn, human-designed questions, where GPT-4-Turbo (Achiam et al., 2023) rates the quality of LLM-generated responses on a ten-point scale. Similarly, AlpacaEval-2 employs GPT-4-Turbo to compare the tested LLMs' responses against GPT-4-Turbo's reference responses and calculates the expected win rate while adjusting for length bias. We repeat the evaluation five times for each setting and report the average score in Table 1 and standard error in Appendix A.

**SFTMix Enhances LLMs' Instruction-Following Capabilities.** As illustrated in Table 1, instruction-tuning with SFTMix consistently outperforms NTP across all metrics in both evaluation benchmarks, regardless of the base LLM or SFT dataset. Notably, SFTMix yields a greater improvement in multi-turn conversational abilities (with an average increase of $0.3$ points) compared to single-turn performance (an average increase of $0.2$ points) in MT-Bench. Across the eight categories in MT-Bench, we observe significant gains in extraction tasks for Llama-3.1-8B, and in writing, coding, and STEM for Mistral-7B-v0.1 (full details in Appendix A). In AlpacaEval-2, the improvement is particularly significant in the length-controlled (LC) win rate, which better aligns with human judgment by adjusting for GPT-4-Turbo's preference for longer responses. While instruction-tuning with the larger, higher-quality UltraChat-200K dataset results in higher overall scores in MT-Bench and raw win rates in AlpacaEval-2, it also produces longer responses, leading to relatively lower LC win rates. Overall, our proposed recipe, SFTMix, enhances instruction-

| LLM | MedQA | MedQA-5 | PubMedQA | MedMCQA | Macro Ave |
|---|---|---|---|---|---|
| MedAlpaca-7B | 38.94 | 33.96 | 57.20 | 34.90 | 41.25 |
| PMC-LLaMA-7B | 27.94 | 21.24 | 54.87 | 24.57 | 32.16 |
| BioMedGPT-LM-7B | 38.62 | 34.72 | 58.27 | 35.57 | 41.80 |
| Meditron-7B | 35.09 | 26.73 | 56.93 | 34.03 | 38.20 |
| BioMistral-7B | 43.86 | 37.58 | 50.13 | 44.14 | 43.93 |
| Llama-3.1-8B | 59.68 | 53.23 | 73.40 | 52.79 | 59.78 |
| + NTP on MedAlpaca-263K | 59.31 | 54.52 | 75.40 | 53.65 | 60.72 |
| or SFTMix on MedAlpaca-263K | **60.88** | **55.38** | **77.80** | **54.15** | **62.05** |
| Mistral-7B-v0.1 | 49.18 | 43.94 | 72.33 | 47.98 | 53.36 |
| + NTP on MedAlpaca-263K | 49.10 | 44.62 | 75.40 | 48.15 | 54.32 |
| or SFTMix on MedAlpaca-263K | **51.77** | **45.72** | **77.40** | **49.03** | **55.98** |

Table 2: Evaluation results on four healthcare-related benchmarks by prior biomedical LLMs and LLMs trained on MedAlpaca-263K using either NTP or SFTMix. We report the mean accuracy (%) over three rounds of three-shot evaluation and bold the scores from SFTMix-tuned LLMs. Standard errors are provided in Appendix A. SFTMix achieves an approximate $1.5\%$ absolute increase in macro-average accuracy compared to NTP for both Llama-3.1-8B and Mistral-7B-v0.1.

following capabilities and text quality across LLMs from different model families and with datasets of varying sizes and quality, surpassing the performance of the conventional NTP paradigm.

**Case Study from MT-Bench.** Figure 3 presents a test example from the extraction category in MT-Bench, showing responses generated by Llama-3.1-8B instruction-tuned on Alpaca-52K using either NTP or SFTMix. In this example, the LLM trained with SFTMix accurately interprets the instructions from both the first- and second-turn queries, correctly extracting the relevant information from the prompt. Notably, it succeeds in answering the second-turn query, which involves calculating the profit margin and performing a ratio comparison. In contrast, the LLM trained with NTP struggles to differentiate between revenue and profit, leading to incorrect responses in both turns.

### 4.2 DOMAIN-SPECIFIC SFT

In healthcare domain-specific SFT, we train two LLMs on a large-scale medical conversation dataset using SFTMix and assess their performance against NTP-tuned counterparts on four healthcare-related question-answering benchmarks.

**Dataset.** MedAlpaca-263K (Han et al., 2023) consists of medical NLP tasks reformatted for instruction tuning and healthcare-related conversations of varying quality crowd-sourced from online platforms, which amounts to a total of 263,257 single-turn interactions. We train Llama-3.1-8B and Mistral-7B-v0.1 on MedAlpaca-263K using either NTP or SFTMix for two epochs and follow the remaining hyperparameter settings described in Section 4.1.

**Evaluation Benchmarks.** We compare the effectiveness of SFTMix to NTP in domain-specific SFT by evaluating the performance of the instruction-tuned LLMs on the following benchmarks:

- MedQA (Jin et al., 2021) includes 1,273 four-choice questions from the US Medical License Exam, testing a broad range of medical knowledge.
- MedQA-5 is the variant of MedQA where each question contains five options.
- PubMedQA (Jin et al., 2019) consists of 500 expert-labeled three-choice questions where the model must predict the answer by reasoning based on a provided PubMed abstract.
- MedMCQA (Pal et al., 2022) comprises 4,183 four-choice questions from the Indian Medical Entrance Exams, covering 2,400 healthcare-related topics across 21 medical subjects.

We adopt the three-shot evaluation setting from Labrak et al. (2024) and report the mean accuracy over three evaluation rounds in Table 2. Standard errors are provided in Appendix A. Additionally, we include prior biomedical LLMs of similar sizes, including MedAlpaca-7B (Han et al., 2023), PMC-LLaMA-7B (Wu et al., 2024), BioMedGPT-LM-7B (Luo et al., 2023), Meditron-7B (Chen et al., 2023), and BioMistral-7B (Labrak et al., 2024), as reference models for comparison.

| Ablation Direction | Loss $\ell = \ell_{\mathrm{NTP}} + \mu\,\ell_{\mathrm{Mixup}}$ | | MT-Bench | | | AlpacaEval-2 | |
| | NTP Loss | Mixup Regularization | ST | MT | Overall | WR | LC WR |
| --- | --- | --- | --- | --- | --- | --- | --- |
| NTP | $\ell_{\mathrm{NTP}}(\text{Full})$ | - | 4.9100 | 3.8150 | 4.3625 | 4.0714 | 8.6528 |
| SFTMix | $\ell_{\mathrm{NTP}}(\text{Full})$ | $\mu\,\ell_{\mathrm{Mixup}}(\text{Conf, Unconf})$ | 5.2125 | 3.9525 | 4.5825 | 4.9031 | 10.3195 |
| Section 5.1 | $\ell_{\mathrm{NTP}}(\text{Full})$ | $\mu\,\ell_{\mathrm{Mixup}}(\text{Conf}', \text{Unconf}')$ | 4.8500 | 4.2625 | 4.5563 | 4.5786 | 10.0483 |
| Section 5.2 | $\ell_{\mathrm{NTP}}(\text{Full})$ | $\ell_{\mathrm{Mixup}}(\text{Conf, Unconf})$ | 4.7050 | 4.1075 | 4.4062 | 3.9450 | 8.2856 |
| | - | $\ell_{\mathrm{Mixup}}(\text{Conf, Unconf})$ | 5.0125 | 4.0000 | 4.5062 | 3.5821 | 7.2964 |
| Section 5.3 | $\ell_{\mathrm{NTP}}(\text{Conf})$ | $\mu\,\ell_{\mathrm{Mixup}}(\text{Conf, Unconf})$ | 4.9775 | 4.1075 | 4.5425 | 4.4496 | 9.7824 |
| | $\ell_{\mathrm{NTP}}(\text{Unconf})$ | $\mu\,\ell_{\mathrm{Mixup}}(\text{Conf, Unconf})$ | 5.1800 | 3.9050 | 4.5425 | 4.2030 | 8.9392 |
| Section 5.4 | $\ell_{\mathrm{NTP}}(\text{High})$ | - | 6.1175 | 5.2575 | 5.6875 | 7.2636 | 11.4490 |
| | $\ell_{\mathrm{NTP}}(\text{High} + \text{Low})$ | - | 5.9000 | 5.1825 | 5.5412 | 6.5871 | 11.9590 |
| | $\ell_{\mathrm{NTP}}(\text{High} + \text{Low})$ | $\mu\,\ell_{\mathrm{Mixup}}(\text{High, Low})$ | 5.8025 | 5.0975 | 5.4500 | 5.9382 | 11.1768 |

Table 3: Ablation studies on variants of SFTMix. We identify four ablation directions and evaluate the instruction-following abilities of Llama-3.1-8B trained with the corresponding loss functions. Conf and Unconf are the confident and unconfident subsets of the Full Alpaca-52K dataset, with confidence derived from the training dynamics of Llama-3.1-8B. Conf′ and Unconf′ are based on Gemma-2B's training dynamics. High refers to Alpaca-GPT4-26K (higher quality), while Low refers to Alpaca-26K (relatively lower quality).

**SFTMix Adapts LLMs to Domain-Specific Tasks More Effectively.** Table 2 demonstrates that SFTMix consistently surpasses NTP across all benchmarks for both backbones. In particular, SFT-Mix leads to a 1.33% absolute improvement (from 60.72% to 62.05%) for Llama-3.1-8B and a 1.66% increase (from 54.32% to 55.98%) for Mistral-7B-v0.1 in macro-average accuracy across the four benchmarks. These models also significantly outperform existing biomedical LLMs across all benchmarks by a clear margin.

## 5 Ablation and Analysis

Following the improvements of SFTMix in instruction-following and domain-specific SFT tasks, we conduct extensive ablation studies to analyze the contribution of each design choice and explore its impact across applications. We identify four ablation directions and summarize the results of training Llama-3.1-8B on Alpaca-52K using variants of SFTMix in Table 3.

### 5.1 Generalizing the Training Dynamics from a Weaker Reference LLM

Inspired by Burns et al. (2024), we investigate the generalization of training dynamics from a weaker reference LLM to a stronger instruction-tuning LLM. Specifically, we identify training dynamics with a weaker reference LLM, Gemma-2B (Team et al., 2024), to divide an SFT dataset into a confident subset (Conf′) and a relatively unconfident subset (Unconf′). These subsets are then fed into the Mixup regularization $\ell_{\mathrm{Mixup}}(\text{Conf}', \text{Unconf}')$ when instruction-tuning Llama-3.1-8B. This alternative approach yields comparable scores on MT-Bench and AlpacaEval-2 to the original SFTMix recipe, which uses the same LLM for both training dynamics and Mixup-based instruction tuning. This finding aligns with the weak-to-strong generalization reported by Burns et al. (2024) and highlights the potential for scaling SFTMix to even stronger LLMs.

### 5.2 Incorporating Mixup as a Regularization is More Effective

Equation 5 uses the Mixup regularization $\ell_{\mathrm{Mixup}}$ to alleviate overfitting and encourage generalization. To fully explore its effect, we experiment with setting $\mu = 1$ in Equation 5 (i.e., $\ell = \ell_{\mathrm{NTP}}(\text{Full}) + \ell_{\mathrm{Mixup}}(\text{Conf, Unconf})$) or only minimizing $\ell_{\mathrm{Mixup}}$ without $\ell_{\mathrm{NTP}}$ (i.e., $\ell = \ell_{\mathrm{Mixup}}(\text{Conf, Unconf})$) during instruction tuning. Table 3 shows that these two variants achieve higher scores on MT-Bench but perform worse on AlpacaEval-2 compared to the baseline of using the conventional NTP method. Furthermore, our SFTMix recipe, which employs $\ell_{\mathrm{Mixup}}$ as a regularization, still outperforms both variants across both benchmarks. This finding highlights the importance of incorporating the tradi-

tional NTP task during SFT and supports the conclusion that Mixup is more effective when used as a regularization alongside the standard cross-entropy loss in LLM instruction tuning.

### 5.3 SFTMix Effectively Utilizes Entire Instruction-Tuning Datasets

As part of our SFTMix recipe, we apply the NTP loss $\ell_{\text{NTP}}$ to the entire SFT dataset $\mathcal{D}$. Here, we consider variants where $\ell_{\text{NTP}}$ is applied selectively to either the confident or relatively unconfident halves of the dataset. Specifically, we experiment with $\ell = \ell_{\text{NTP}}(\text{Conf}) + \mu \, \ell_{\text{Mixup}}(\text{Conf}, \text{Unconf})$ and $\ell = \ell_{\text{NTP}}(\text{Unconf}) + \mu \, \ell_{\text{Mixup}}(\text{Conf}, \text{Unconf})$. As shown in Table 3, while both variants achieve the same overall score on MT-Bench, the variant applying $\ell_{\text{NTP}}$ to the confident subset (Conf) performs better on AlpacaEval-2. Notably, both variants—where $\ell_{\text{NTP}}$ is applied to only half the dataset—outperform the baseline where $\ell_{\text{NTP}}$ is applied to the entire dataset. We attribute this improvement to the impact introduced by our Mixup regularization $\ell_{\text{Mixup}}$. Nevertheless, our SFTMix recipe, which leverages the full dataset for NTP, outperforms both variants, demonstrating its ability to effectively utilize a larger set of potentially lower-quality training examples during instruction tuning.

### 5.4 Training Dynamics Are Crucial for Performing Mixup

Building on the previous ablation study, which suggests the possibility of generalizing training dynamics from a weaker LLM in our SFTMix recipe, we explore whether we can directly substitute training dynamics with known data quality. To test this hypothesis, we replace half of the original responses in Alpaca-52K with higher-quality GPT-4-generated versions, forming Alpaca-GPT4-26K (High) (Peng et al., 2023), while referring to the remaining original responses as Alpaca-26K (Low). We then train Llama-3.1-8B using three approaches: NTP on Alpaca-GPT4-26K ($\ell_{\text{NTP}}(\text{High})$), NTP on its combination with Alpaca-26K ($\ell_{\text{NTP}}(\text{High} + \text{Low})$), and, in the final approach, the addition of the Mixup-based regularizer between Alpaca-GPT4-26K and Alpaca-26K ($\ell_{\text{Mixup}}(\text{High}, \text{Low})$). The use of higher-quality responses from GPT-4 indeed enhances instruction-tuning performance on both MT-Bench and AlpacaEval-2, as shown in Table 3. However, simply applying Mixup between two datasets of varying quality does not necessarily improve performance further, as indicated by the drop in the overall MT-Bench score from $5.5412$ to $5.4500$ and the

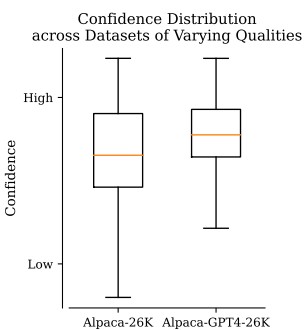

Figure 4: Distribution of confidence in datasets of varying qualities by Llama-3.1-8B.

length-controlled win rate in AlpacaEval-2 from $11.9590$ to $11.1768$. To investigate this observation, we plot the LLM's confidence distributions for both datasets in Figure 4. The substantial overlap in confidence distributions suggests that data quality does not necessarily correlate with training dynamics-based confidence. This highlights the importance of training dynamics in determining the model-specific role of data points, which is crucial for effectively applying our SFTMix recipe.

## 6 Conclusion

In this paper, we propose SFTMix, a novel recipe for LLM instruction tuning. We observe that LLMs exhibit uneven confidence distributions across the semantic representation space. Based on this motivation, we utilize training dynamics to identify data subsets of varying confidence levels and incorporate a Mixup-based regularization. In this way, we aim to mitigate overfitting on the confident subset while propagating supervision signals to promote the generalization of the relatively unconfident subset. Extensive empirical results in both instruction-following and domain-specific SFT tasks demonstrate the effectiveness of SFTMix over the conventional NTP paradigm across different LLM families and SFT data scales. Comprehensive ablation studies further substantiate the contribution of SFTMix's design choices, highlighting its versatility in consistently enhancing performance across different LLMs and datasets in broader NLP applications. Due to computational constraints, we did not apply SFTMix to LLM pre-training or instruction-tune larger LLMs using this recipe. Integrating SFTMix with parameter-efficient pre-training and fine-tuning methods (Hu et al., 2022; Dettmers et al., 2024) is another promising direction for future work.

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

# A  EXPERIMENT DETAILS

In Section 4, we assess the effectiveness of SFTMix against NTP in both instruction-following and domain-specific SFT tasks. Here, we report the detailed experiment results with standard errors.

| Dataset | Instruction-Tuning LLM | Recipe | Single-Turn | MT-Bench Multi-Turn | Overall | AlpacaEval-2 Win Rate | LC Win Rate |
|---|---|---|---|---|---|---|---|
| Alpaca-52K | Llama-3.1-8B | NTP | $4.9100 \pm 0.06$ | $3.8150 \pm 0.06$ | $4.3625 \pm 0.05$ | $4.0714 \pm 0.14$ | $8.6528 \pm 0.19$ |
| | | SFTMix | $\mathbf{5.2125} \pm 0.03$ | $\mathbf{3.9525} \pm 0.07$ | $\mathbf{4.5825} \pm 0.03$ | $\mathbf{4.9031} \pm 0.21$ | $\mathbf{10.3195} \pm 0.04$ |
| | Mistral-7B-v0.1 | NTP | $5.1650 \pm 0.08$ | $4.0675 \pm 0.13$ | $4.6163 \pm 0.05$ | $4.3560 \pm 0.15$ | $9.1759 \pm 0.17$ |
| | | SFTMix | $\mathbf{5.2775} \pm 0.03$ | $\mathbf{4.5425} \pm 0.10$ | $\mathbf{4.9100} \pm 0.05$ | $\mathbf{4.5386} \pm 0.18$ | $\mathbf{9.4994} \pm 0.28$ |
| UltraChat-200K | Llama-3.1-8B | NTP | $6.1875 \pm 0.04$ | $5.0125 \pm 0.02$ | $5.6000 \pm 0.02$ | $5.0665 \pm 0.12$ | $8.4505 \pm 0.08$ |
| | | SFTMix | $\mathbf{6.2750} \pm 0.07$ | $\mathbf{5.3500} \pm 0.04$ | $\mathbf{5.8125} \pm 0.02$ | $\mathbf{5.1149} \pm 0.16$ | $\mathbf{9.3810} \pm 0.25$ |
| | Mistral-7B-v0.1 | NTP | $5.7625 \pm 0.03$ | $4.6938 \pm 0.01$ | $5.2281 \pm 0.01$ | $4.4899 \pm 0.18$ | $7.7732 \pm 0.21$ |
| | | SFTMix | $\mathbf{5.9813} \pm 0.08$ | $\mathbf{4.8813} \pm 0.03$ | $\mathbf{5.4313} \pm 0.03$ | $\mathbf{4.6117} \pm 0.04$ | $\mathbf{8.7650} \pm 0.21$ |

Table 4: Evaluation of instruction-following capabilities of LLMs trained with NTP or SFTMix. We report the average score and standard errors based on five rounds of evaluation, with the scores from the best-performing instruction-tuning recipe in bold. SFTMix outperforms NTP on both MT-Bench and AlpacaEval-2, irrespective of LLM families and SFT data sizes.

| LLM | MedQA | MedQA-5 | PubMedQA | MedMCQA | Macro Ave |
|---|---|---|---|---|---|
| MedAlpaca-7B | $38.94 \pm 0.37$ | $33.96 \pm 0.26$ | $57.20 \pm 0.71$ | $34.90 \pm 0.39$ | $41.25 \pm 0.40$ |
| PMC-LLaMA-7B | $27.94 \pm 0.65$ | $21.24 \pm 0.56$ | $54.87 \pm 0.62$ | $24.57 \pm 0.27$ | $32.16 \pm 0.40$ |
| BioMedGPT-LM-7B | $38.62 \pm 1.51$ | $34.72 \pm 0.46$ | $58.27 \pm 0.25$ | $35.57 \pm 0.52$ | $41.80 \pm 0.52$ |
| Meditron-7B | $35.09 \pm 0.64$ | $26.73 \pm 0.19$ | $56.93 \pm 1.27$ | $34.03 \pm 0.36$ | $38.20 \pm 0.39$ |
| BioMistral-7B | $43.86 \pm 0.33$ | $37.58 \pm 0.62$ | $50.13 \pm 0.66$ | $44.14 \pm 0.33$ | $43.93 \pm 0.27$ |
| Llama-3.1-8B | $59.68 \pm 0.33$ | $53.23 \pm 0.21$ | $73.40 \pm 0.86$ | $52.79 \pm 0.01$ | $59.78 \pm 0.51$ |
| + NTP on MedAlpaca-263K | $59.31 \pm 0.56$ | $54.52 \pm 0.21$ | $75.40 \pm 0.57$ | $53.65 \pm 0.18$ | $60.72 \pm 0.08$ |
| or SFTMix on MedAlpaca-263K | $\mathbf{60.88} \pm 0.29$ | $\mathbf{55.38} \pm 0.13$ | $\mathbf{77.80} \pm 0.16$ | $\mathbf{54.15} \pm 0.11$ | $\mathbf{62.05} \pm 0.24$ |
| Mistral-7B-v0.1 | $49.18 \pm 0.30$ | $43.94 \pm 0.23$ | $72.33 \pm 0.20$ | $47.98 \pm 0.22$ | $53.36 \pm 0.43$ |
| + NTP on MedAlpaca-263K | $49.10 \pm 0.33$ | $44.62 \pm 0.39$ | $75.40 \pm 0.68$ | $48.15 \pm 0.11$ | $54.32 \pm 0.62$ |
| or SFTMix on MedAlpaca-263K | $\mathbf{51.77} \pm 0.51$ | $\mathbf{45.72} \pm 0.44$ | $\mathbf{77.40} \pm 0.28$ | $\mathbf{49.03} \pm 0.22$ | $\mathbf{55.98} \pm 0.31$ |

Table 5: Evaluation results on four healthcare-related benchmarks by prior biomedical LLMs and LLMs instruction-tuned on MedAlpaca-263K using either NTP or SFTMix. We report the mean accuracy (%) and standard errors over three rounds of three-shot evaluation and bold the scores from SFTMix-tuned LLMs. FTMix achieves an approximate 1.5% absolute increase in macro-average performance compared to NTP for both Llama-3.1-8B and Mistral-7B-v0.1.

| Instruction-Tuning LLM | Recipe | Writing | Roleplay | Reasoning | Math | MT-Bench Coding | Extraction | STEM | Humanities | Overall |
|---|---|---|---|---|---|---|---|---|---|---|
| Llama-3.1-8B | NTP | 6.79 | 4.93 | 2.94 | 1.82 | **2.63** | 5.96 | 4.46 | 5.37 | 4.36 |
| | SFTMix | **6.81** | **5.15** | **3.12** | **1.85** | **2.63** | **6.80** | **4.77** | **5.53** | **4.58** |
| Mistral-7B-v0.1 | NTP | 6.23 | 5.20 | **4.81** | 1.21 | 2.63 | 6.82 | 4.44 | 5.59 | 4.62 |
| | SFTMix | **7.08** | **5.42** | 4.51 | **1.27** | **3.45** | **6.90** | **4.97** | **5.68** | **4.91** |

Table 6: Categorical evaluation of instruction-following capabilities of LLMs trained on Alpaca-52K using NTP or SFTMix. We report the average score in each of the eight categories in MT-Bench based on five rounds of evaluation, with the scores from the best-performing instruction-tuning recipe in bold. By using SFTMix, we observe significant gains in extraction tasks for Llama-3.1-8B, and in writing, coding, and STEM for Mistral-7B-v0.1.

