# OpenReview forum: "SFTMix: Elevating Language Model Instruction Tuning with Mixup Recipe"
_ICLR.cc/2025/Conference — ICLR 2025 Conference Withdrawn Submission_

### Official Review · Reviewer_fsBM · 2024-10-20

**Soundness:** 1
**Presentation:** 2
**Contribution:** 1
**Rating:** 3
**Confidence:** 4

**Summary:**

This paper proposes SFTMix to promote conventional instruction-tuning paradigm for LLMs. While conventional instruction-tuning uses Next-Token Prediction Loss and Well-Curated Datasets, SFTMix claims to propose a new loss function leveraging the naive datasets, without the need to complex data selection process.
The algorithm:
1. Firstly, divides the trainset into confident/unconfident part based on a ppl-based confidence-score. The score is calculated by some reference checkpoints of the waiting-to-trained LLMs;.
2. Secondly, adds a regularization term into the loss function to regularize the overfitting of confident examples and enhance the fitting of unconfident examples. The regularization term is calculated between linearly-interpolated representations and labels from confident and unconfident examples.

Through experiments on datasets/benchmarks in both general and specific domains, SFTMix shows advantage to SFT.

**Strengths:**

- SFTMix further explores the probability to enhance Instruction Tuning process by leveraging positive and negative answers for different instructions. In contrast, RLHF and DPO need the positive and negative answers for the same instruction to compute the contrastive loss.
- The tested models, datasets, and benchmarks are various. Thus, the results well support the advantage of SFT-Mix to SFT.

**Weaknesses:**

- **(Perhaps) Over-claimed contribution**

  In SFTMix's optimization progress, it seems that **the main contribution to vectorial gradient map is still from Next-Token Prediction loss, but not from its proposed regulatization term**. The reasons are as follows:
  - Firstly, the representation $Z$ of both confident and unconfident examples are extracted from the last layer, which means that the whole vectorial gradient map have been established and fixed during the forward pass. The regularization term will only influence the scalar weight of the gradient brought by each token. Therefore, **the loss function is still in the format of Next-Token Prediction loss, but only a token-level scalar reweighting trick is added on** (scalar weight may be less important than vectorial direction).
  - Secondly, it is unclear that, when computing the mixup-regularization-term, whether the "require_grad" is set to True or False. If it is set to False, the regularization term will only influence the sentence-level scalar gradient weight, but not influence the token-level weight at all. **If so, SFTMix will be a method totally based on Next-Token Prediction loss, but only a sentence-level reweighting trick is added on.**
- **Inadequate experiments for SFTMix's core motivation**
  - This paper claims that SFTMix will regularize the overfitting of confident examples and enhance the fitting of unconfident examples. However, the paper only visualize the semantic representation space of before-trained LLMs in Figure 2, but not for the trained ones by SFT/SFTMix. Thus, whether their method works as their claim is unclear.
  - This paper claims that SFTMix will eliminate the complex process of data selection. However, **they only conduct experiments on well-curated datasets such as UltraChat-200K (while the original UltraChat dataset are much larger than 200K)**. Thus, it is inadequate to demonstrate SFTMix's effectiveness on non-curated datasets. Also, **it is unclear whether SFTMix will still work when there is a lot of "dirty data" bringing "dirty gradient", as these data tends to be unconfident for LLMs and SFTMix will give them larger weights.**
- **Inadequate related works and baseline**

  From my view, the three main parts of SFTMix algorithm are: ① Confidence/importance-based data selection [1, 2]; ② Token-level distribution noise-addition/regularization [3, 4, 5]; ③*Alignment using Instruction-irrelavant negative example [6]. **These methods have been well-explored before, but this paper did not consider any of these works as related work or baselines** (only SFT is chosen for comparison).

  [1] Data Selection for Language Models via Importance Resampling.

  [2] LESS: Selecting Influential Data for Targeted Instruction Tuning.

  [3] NEFTune: Noisy Embeddings Improve Instruction Finetuning.

  [4] Token-level Direct Preference Optimization.

  [5] Intuitive Fine-Tuning: Towards Simplifying Alignment into a Single Process.

  [6] KTO: Model Alignment as Prospect Theoretic Optimization.

**Questions:**

- When computing the mixup-regularization-term, whether the "require_grad" is set to True or False? And how do you clarify the differences between SFT and SFTMix, especially from the view of gradient map? (e.g. ① Detailed formula proof; ② Quantitative gradient contributions analysis; ③ or other evidence.)
- How do you clarify SFTMix's influence on fitting confident/unconfident examples when comparing with baselines? (e.g. ① Visualizations of the semantic representation space after SFT/SFTMix; ② Statics of confidence scores change after SFT/SFTMix; ③ or other evidence.)
- How do you estimate and clarify SFTMix's performance on truly-uncurated/noisy datasets (but not pre-curated datasets in this paper)?
- How do you compare SFTMix with the related methods mentioned above? Especially, what is the novel contribution and main advantages of SFTMix compare with these methods?

Welcome for further explanation for the the weaknesses and questions mentioned above, and I might reconsider my review after that.

---

### Official Review · Reviewer_Xaok · 2024-11-01

**Soundness:** 2
**Presentation:** 2
**Contribution:** 2
**Rating:** 3
**Confidence:** 4

**Summary:**

The authors propose SFTMix, a novel recipe for language model instruction tuning that leverages training dynamics to identify data subsets with varying confidence levels and applies a Mixup-based regularization to improve learning. This approach aims to enhance instruction-tuning performance beyond the traditional Next-Token Prediction (NTP) paradigm without relying on well-curated datasets.

**Strengths:**

1.	SFTMix introduces a unique approach that combines insights from training dynamics and Mixup regularization, offering a fresh perspective on instruction tuning.
2.	The paper presents compelling evidence of SFTMix’s effectiveness across various instruction-following and domain-specific tasks, showcasing consistent improvements over the NTP baseline.

**Weaknesses:**

1.	The author raises the issue of the cost of instruction dataset filtering, but to my knowledge, even simple instruction filtering using models such as GPT4 can achieve good results [1]. However, the author's method still requires pre training of the dataset, which undoubtedly increases additional costs.
2.	In my view, the core of the method proposed by the author still falls under a type of regularization technique. But as far as I know, on the alpaca dataset, simple LoRA training can also achieve results comparable to or even better than full-parameter NTP [2]. Therefore, the question arises whether the author's complex fine-tuning method has practical significance.
3.	The performance of the method in the paper is significantly affected by hyperparameter selection, but unfortunately, the author does not explore this aspect in the experiment.
4.	The author refers to “high confidence” several times throughout the paper; however, the experimental details do not clarify what threshold is considered “high confidence.” It necessitates further elucidation from the author regarding whether the setting of this parameter influences the experimental outcomes.

[1] L. Chen, S. Li, J. Yan, H. Wang, K. Gunaratna, V. Yadav, Z. Tang, V. Srinivasan, T. Zhou, 306 H. Huang, and H. Jin. Alpagasus: Training a better alpaca with fewer data. In The Twelfth 307 International Conference on Learning Representations, ICLR 2024, Vienna, Austria, May 7-11, 308 2024.
[2] S. Ghosh, C. K. R. Evuru, S. Kumar, R. S., D. Aneja, Z. Jin, R. Duraiswami, and D. Manocha. 336 A closer look at the limitations of instruction tuning. In Forty-first International Conference 337 on Machine Learning, ICML 2024, Vienna, Austria, July 21-27, 2024

**Questions:**

1.	The author raises the issue of the cost of instruction dataset filtering, but to my knowledge, even simple instruction filtering using models such as GPT4 can achieve good results [1]. However, the author's method still requires pre training of the dataset, which undoubtedly increases additional costs.
2.	In my view, the core of the method proposed by the author still falls under a type of regularization technique. But as far as I know, on the alpaca dataset, simple LoRA training can also achieve results comparable to or even better than full-parameter NTP [2]. Therefore, the question arises whether the author's complex fine-tuning method has practical significance.
3.	The performance of the method in the paper is significantly affected by hyperparameter selection, but unfortunately, the author does not explore this aspect in the experiment.
4.	The author refers to “high confidence” several times throughout the paper; however, the experimental details do not clarify what threshold is considered “high confidence.” It necessitates further elucidation from the author regarding whether the setting of this parameter influences the experimental outcomes.

[1] L. Chen, S. Li, J. Yan, H. Wang, K. Gunaratna, V. Yadav, Z. Tang, V. Srinivasan, T. Zhou, 306 H. Huang, and H. Jin. Alpagasus: Training a better alpaca with fewer data. In The Twelfth 307 International Conference on Learning Representations, ICLR 2024, Vienna, Austria, May 7-11, 308 2024.
[2] S. Ghosh, C. K. R. Evuru, S. Kumar, R. S., D. Aneja, Z. Jin, R. Duraiswami, and D. Manocha. 336 A closer look at the limitations of instruction tuning. In Forty-first International Conference 337 on Machine Learning, ICML 2024, Vienna, Austria, July 21-27, 2024

---

### Official Review · Reviewer_kFed · 2024-11-02

**Soundness:** 3
**Presentation:** 3
**Contribution:** 2
**Rating:** 3
**Confidence:** 4

**Summary:**

This paper proposes SFTMix, a new method for the data mixture for supervised finetuning. The method trains a reference model to predict the confidence of the data and uses a mixup method to leverage different data for the improvement in downstream tasks and the mitigation of overfitting to confident examples. Experimental results demonstrate the effectiveness of this method. Comprehensive ablation studies further validate the robustness of the proposed design choices.

**Strengths:**

1. The proposed method is tackling an important and common problem in SFT, which is the data mixture of data of different qualities. The methos does not rely on well-curated dataset but leverages the confidence scores for a good data mixture to improve performance and mitigate overfitting to confident examples.
2. It seems that this method can be adaptive to different LLMs and different datasets, which demonstrates its robustness.

**Weaknesses:**

1. This method may introduce additional complexity with the mixup framework, including the training dynamics and mixup regularization. Furthermore, this article should provide more discussion about the choice of hyperparameters, especially the hyperpameter $\lambda$ for linear interpolation and $\mu$ for mixup regularization.
2. The training of reference model determines the confidence scores, which is the core of the method. The article should discuss why the training dynamics can be trusted as the confidence in its quality. The dependence on training a reference model may introduce more errors if the training is subpar. Please provide more empirical evidence or insightful discussion for the significance of using reference model to determine the confidence scores. Also, if this study can provide some alternatives, such as data quality judger to provide quality scores for data mixture, for the comparison, that will be helpful for readers to understand the significance.
3. Conducting experiments on MT-Bench and Alpaca-Eval is still far from enough. To further conduct automatic evaluation for the human preference alignment and instruction following, it is suggested to conduct experiments on datasets like Arena-Hard, IF-Eval, etc. Also, it is suggested to provide human evaluation with sophisticated design in order to exactly evaluating human preference alignment.
4. It is doubtful whether the method can be generalized to larger models, such as 30B to 70B LLMs, or larger-scale datasets, such as Open-Hermes,  to build a significantly better model.

**Questions:**

1. How to evaluate the quality of the reference model? Its quality is deterministic to the whole method.
2. Is it possible to conduct experiments on larger models?

---

### Official Review · Reviewer_jBVB · 2024-11-03

**Soundness:** 4
**Presentation:** 3
**Contribution:** 3
**Rating:** 5
**Confidence:** 4

**Summary:**

The paper introduces SFTMix, which uses Mixup-based regularization to interpolate the representations of training data with different confidence levels. This approach mitigates overfitting on high-confidence examples while propagating supervision signals to enhance learning on lower-confidence ones. The authors demonstrate the effectiveness of SFTMix through extensive experiments in instruction-following tasks and the healthcare domain.

**Strengths:**

The authors establish a connection between confidence levels and semantic representations from a decision boundary perspective, and propose SFTMix to address overfitting in high-confidence regions while enhancing generalization in low-confidence areas. The proposed method is easy to follow and demonstrates good generalization across different tasks, models, and domains.

**Weaknesses:**

1. Although Figure 2(a) does show a clear separation between high-confidence and low-confidence data, I still recommend the authors provide a more in-depth analysis to clarify the relationship between confidence and semantic representation. Either intuitive or empirical insights would help readers understand the significance of the problem being addressed.

    For example:

    - More case studies (e.g., cases at varying distances from the boundary), statistical analysis (task type, vocabulary distribution, etc.)

    - Questions such as "What is the primary reason for the differences in semantic distributions? Is it due to the varying semantic complexity or difficulty of different instruction tasks(and generated responses), or is it more related to the confidence levels?" could be answered.


2. While SFTMix is generally simple and effective, its application requires training a reference LLM on the entire dataset to determine data confidence levels. This process ultimately leads to more than double the computational cost, which isn't friendly for large-scale SFT (say, with millions of training examples). I'm curious whether using the training dynamics of only a subset of data to determine confidence levels could achieve similar performance improvements.

**Questions:**

1. Possible typo in Figure 2(b) unconfident example: "**Fine**(Find?) the most suitable name..."

2. How are instances selected for mixup? Is it a random pairing of high-confidence and low-confidence data?

3. I'm curious about the performance of models trained separately on only Conf and Unconf subsets. This would help us understand how data with different confidence levels impacts performance.

---

### Note · Authors · 2024-11-24

I have read and agree with the venue's withdrawal policy on behalf of myself and my co-authors.